# High pulse pressure is associated with an increased risk of prediabetes in hypertensive individuals: A retrospective study based on an adult Chinese population

**Zhanxing Wu**[1◉], **Zhongqing Chen**[1◉], **Wenfei Zeng**[2◉], **Ganggang Peng**[1]*, **Zhenhua Huang**[1]*

1 Department of Emergency, Shenzhen Second People's Hospital, Shenzhen, Guangdong, China,
2 Department of Anesthesiology, Hunan Provincial People's Hospital, The First Affiliated Hospital of Hunan Normal University, Changsha, China

◉ These authors contributed equally to this work.
* gyy405@126.com (GP); huangzhh12306@163.com (ZH)

## Abstract

**Data Availability Statement:** All relevant data are within the paper and its Supporting Information files.

### Objectives

The pulse pressure (PP) is an important factor influencing the outcomes of diabetes. However, the relationship between the PP and prediabetes has been rarely studied and how this association might be impacted by hypertension is not clear.

### Methods

In this study, we retrospectively included 184,252 adults from 32 regions in China, spanning from 2010 to 2016. Cox regression and sensitivity analysis were used to examine the relationship between PP and prediabetes. For the hypertensive population, Cox proportional hazards regression and smooth curve fitting were performed to explore the non-linear relationship between PP and prediabetes. A two-stage Cox proportional hazards regression model was used to determine the inflection point of PP in relation to the risk of prediabetes.

### Results

After adjusting for confounding factors, we found a positive association between PP and prediabetes (HR: 1.11, 95% CI: 1.03–1.19, P = 0.0045). However, we observed that this relationship was not significant in the normal blood pressure group (HR: 1.06, 95% CI: 0.94–1.18, P = 0.3529). We also found a non-linear relationship between PP and the risk of prediabetes in hypertensive individuals. The inflection point of PP was 31 mmHg. When PP ≥ 31 mmHg, there was a positive association with the risk of prediabetes (HR: 1.22, 95% CI: 1.04–2.08, P <0.0001). Conversely, when PP < 31 mmHg, this association was not significant (HR: 0.83, 95% CI: 0.59–1.15, P = 0.2646).

**Funding:** The author(s) received no specific funding for this work.

**Competing interests:** The authors declare that they have no known competing financial interests or personal relationships that could have appeared to influence the work reported in this paper.

## Conclusions

This study suggests a non-linear relationship between PP and the risk of prediabetes in hypertensive individuals. Maintaining PP within 31 mmHg is crucial for preventing the occurrence of prediabetes.

## Introduction

Blood pressure is a crucial physiological indicator that plays a significant role in assessing cardiovascular health and preventing related diseases [1]. Maintaining normal blood pressure is essential for safeguarding the heart, brain, and other vital organs from damage. Hypertension increases the risk of severe health issues, including cardiovascular diseases, stroke, and kidney diseases [2]. The pulse pressure (PP) refers to the difference between the systolic blood pressure (SBP) and diastolic blood pressure (DBP), and it provides more information about the functional status and vascular elasticity of the cardiovascular system compared to the SBP or DBP alone. Therefore, it is closely associated with many cardiovascular diseases. Arterial stiffness leads to an increased forward wave amplitude and earlier arrival of the reflected wave, resulting in elevated pulse pressure. This indicates a close correlation between PP and arterial stiffness [3]. In addition, PP is associated with heart attacks, heart failure and strokes [4–6].

Prediabetes, is a condition characterized by blood glucose levels that are higher than the normal range but below the diagnostic criteria for diabetes. The identification of prediabetic individuals is based on impaired fasting glucose, impaired glucose tolerance, and glycated hemoglobin [7]. According to the American Diabetes Association, prediabetes is defined as a fasting blood glucose level between 5.6–6.9 mmol/L [7]. Globally, prediabetes is widely prevalent and has become a global public health issue. According to the International Diabetes Federation, approximately 415 million adults worldwide have prediabetes. It is estimated that up to 70% of individuals with prediabetes will progress to diabetes within 10 years without intervention [8]. In the US, the situation is more severe, with about 10% of individuals with prediabetes transitioning to diabetes each year [9]. Thus, early identification and intervention for prediabetes may play a crucial role in preventing the progression to diabetes. In addition to traditional blood glucose markers, researchers have recently been exploring novel indicators for the early detection of prediabetes. These include inflammatory markers (such as C-reactive protein), metabolic syndrome indicators, vitamin D and insulin resistance indicators (such as the insulin resistance index and insulin sensitivity index), which are believed to be potentially associated with the risk of prediabetes [10–13]. However, these new indicators often require invasive procedures and have not yet been widely adopted in clinical practice.

Recent studies have suggested that a high PP was associated with the occurrence and development of diabetes. A study involving 209,635 participants found that a high PP was associated with an increased incidence of diabetes [14]. However, the available data on the relationship between the PP and prediabetes in the Chinese population is relatively scarce and how this association might be impacted by hypertension is not clear. Therefore, we conducted a study to explore the relationship between PP and the risk of prediabetes, as well as to investigate the impact of hypertension on this association. Through this study, we aim to gain a better understanding of the connection between PP and prediabetes and provide essential insights for preventing the onset of prediabetes.

## Methods

### Data source

The Dryad Digital Repository is an open-access data repository designed to provide researchers with a reliable platform for storing, sharing, and discovering scientific research data. Researchers are allowed to utilize the downloaded data for analysis, processing, visualization, or other scientific research activities. Users have the freedom to modify, integrate, or transform the data as needed. We obtained the raw data from the Dryad data repository (dataset: https://datadryad.org/stash/dataset/https://doi:10.5061%2Fdryad.ft8750v) and conducted a secondary analysis of a medical examination program using the publicly available data provided by Ying Chen et al. [15]. The original study adhered to the guidelines outlined in the Declaration of Helsinki and received approval from the Rich Healthcare Group Review Board. Furthermore, the Rich Healthcare Group Review Board has waived the requirement for informed consent for the current retrospective study.

### Study population

The original researchers retrospectively obtained data for the Dryad Digital Repository from a computerized database developed by the Rich Healthcare Group in China. This database includes the medical records of individuals who underwent health check-ups between 2010 and 2016, covering 32 regions and 11 cities in China. The original study and data collection were approved by the Rich Healthcare Group Review Board. Because the present study was a secondary analysis of retrospective data, it did not require informed consent or approval from an institutional ethics committee.

The original study comprised 685,277 participants aged 20 and older who underwent at least two health examinations. Participants with follow-up fasting plasma glucose levels between 6.1 and 6.9 mmol/l and no new reports of diabetes were included. Initial exclusions at baseline were based on the following criteria: (1) insufficient information on weight, height, and gender; (2) extreme BMI values ($<15$ kg/m$^2$ or $>55$ kg/m$^2$); (3) visit intervals $<2$ years; (4) absence of fasting plasma glucose value; (5) participants diagnosed with diabetes at baseline and those with undefined diabetes status at follow-up. After applying these exclusion criteria, 211,833 participants remained in the original study [15].

In the subsequent analysis, 27,631 participants were further excluded, including those with: 1) missing available FPG values during follow-up, 2) missing available DBP or SBP values, 3) individuals with FPG$\geq$5.6 mmol/l at baseline, 4) individuals with FPG $> 6.9$ mmol/l during follow-up, and 5) individuals diagnosed as undefined with diabetes during follow-up. A total of 184,252 participants were finally included in this study. The participant selection process is illustrated in Fig 1.

### Variables

**Pulse pressure (PP).** The PP refers to the difference between the SBP and DBP and was recorded as a continuous variable. The PP was defined as PP = SBP-DBP (mmHg) [16].

**Data collection.** Demographic information, including the patient age, SBP, DBP, height, and weight, was collected. BMI is calculated based on height and weight. Regarding the demographic information and related measurements such as blood pressure, the trained staff underwent retraining to ensure consistency in their work. Additionally, measurements were taken for the FPG, serum creatinine (Scr), triglyceride (TG), total cholesterol (TC), blood urea nitrogen (BUN), alanine aminotransferase (ALT), low-density lipoprotein cholesterol (LDL-C) and high-density lipoprotein cholesterol (HDL-C) levels. All tests were conducted in the same type

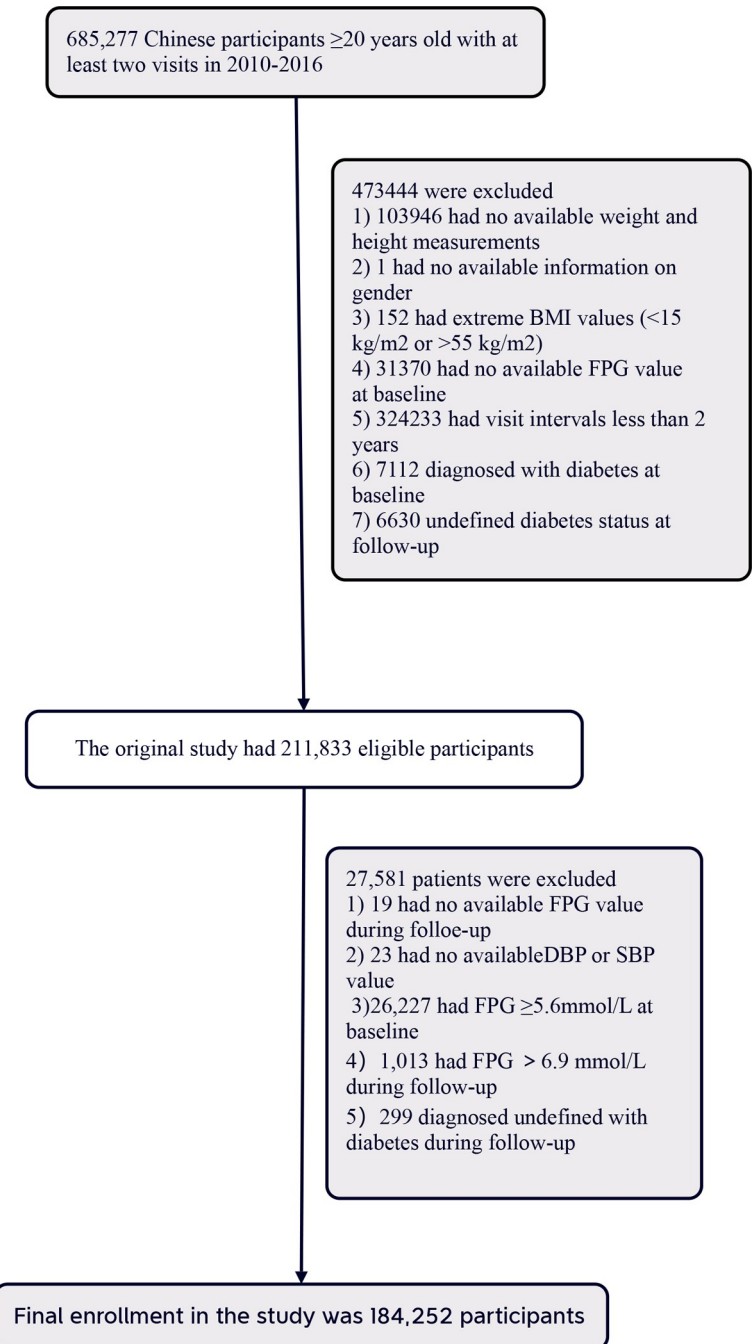

**Fig 1. Flowchart of study participants.**

of laboratory. The PP served as the independent variable, while the occurrence of prediabetes during the follow-up period was the dependent variable.

**Definition.** Prediabetes was defined as having impaired fasting glucose levels (FPG: 5.6–6.9 mmol/l) [7].

**Outcome measures.** The occurrence of prediabetes was our outcome variable. Prediabetes was defined based on the FPG level at the follow-up evaluation, and the absence of self-

reported incident diabetes during the follow-up period. Prediabetes was present for a FPG of 5.6–6.9 mmol/l.

## Statistical analysis

Individuals were divided into hypertensive and normotensive groups based on the guidelines published by the American Heart Association (AHA) and the American College of Cardiology (ACC) in 2017 [1]. Hypertension was defined as a SBP ≥130 mmHg or a DBP ≥80 mmHg. The subjects were divided into four groups based on the quartiles of the PP: Q1 ≤ 35 mmHg; 36< Q2 ≤42 mmHg; 43< Q3 ≤50 mmHg; and Q4>51 mmHg. The mean with standard deviation was used to represent continuous variables that followed a normal distribution, while a median with interquartile range was used to represent continuous variables with a skewed distribution. Categorical variables were presented as percentages of different groups. A one-way ANOVA or Kruskal-Wallis tests was conducted to compare continuous variables, and the chi-squared test was used to compare categorical variables. Incidence rates were calculated using person-years and cumulative incidence rates. Survival and cumulative event rates were analyzed using the Kaplan-Meier method. The log-rank test was performed to assess the hazard ratios (HR) of adverse events.

The Impact of each variable on the risk of prediabetes was assessed using the univariate Cox regression method. Furthermore, the precise relationship between the PP and the risk of prediabetes was analyzed using multivariate Cox regression analysis. Additionally, we used non-adjusted, minimally-adjusted, and fully-adjusted models to further investigate the association between the PP and the risk of developing prediabetes. The model was as follows: (i) Crude model: (unadjusted covariates); (ii) minimum adjustment model (Model I: adjusted for age and sex); (iii) fully adjusted model (Model II: adjusted age, sex, BMI, FPG, TC, BUN, ALT, LDL-c, Scr, TG, HDL-c, family history of diabetes, drinking status, and smoking status).

To ensure the robustness of our findings, we conducted several sensitivity analyses. Empower Stats (X&Y Solutions, Inc., Boston, MA, http://www.empowerstats.com) was used for all analyses. All p values were two-tailed and a p value <0.05 was considered statistically significant.

## Ethics statement

The original study adhered to the guidelines outlined in the Declaration of Helsinki and received approval from the Rich Healthcare Group Review Board. Furthermore, the Rich Healthcare Group Review Board has waived the requirement for informed consent for the current retrospective study. In this study, we anonymize participants. All aspects of the study were conducted in accordance with the relevant provisions of the Declaration of Helsinki.

## Results

### Baseline characteristics of participants

Table 1 shows the demographic and clinical characteristics of the study participants between the normotensive and hypertensive group. A total of 184,252 individuals were included. Their mean age was 41.02 ± 12.10 years old. Slightly more than half (53.07%) of the individuals were men, and 20743 (11.26%) individuals eventually were diagnosed with prediabetes after a follow-up of an average of 3.14 years. Compared to the normotensive group, the hypertensive group had higher age, height, weight, BMI, PP, TC, TG, LDL-c, ALT, BUN, Scr, FPG, SBP, DBP and risk of prediabetes during the follow-up. In addition, the hypertensive group had higher rates of males, smokers, and drinkers. In comparison to the hypertensive group, the

**Table 1. Clinical and biochemical characteristics between the normotensive and hypertensive group.**

| Variable | All participants (n = 184,252) | Normotensive group (n = 121,237) | Hypertensive group (n = 63,015) | P-value |
|---|---|---|---|---|
| Age (years) | 41.02 ± 12.10 | 38.91 ± 10.38 | 45.08 ± 13.97 | <0.001 |
| Height (cm) | 166.39 ± 8.32 | 165.77 ± 8.19 | 167.59 ± 8.44 | <0.001 |
| Weight (kg) | 63.97 ± 12.04 | 61.59 ± 11.15 | 68.55 ± 12.37 | <0.001 |
| BMI (kg/m2) | 22.99 ± 3.27 | 22.31 ± 2.98 | 24.31 ± 3.38 | <0.001 |
| SBP (mmHg) | 117.83 ± 15.81 | 109.89 ± 10.17 | 133.09 ± 13.34 | <0.001 |
| DBP (mmHg) | 73.53 ± 10.60 | 68.01 ± 6.55 | 84.15 ± 8.62 | <0.001 |
| PP (mmHg) | 44.30 ± 11.28 | 41.89 ± 8.61 | 48.94 ± 14.02 | <0.001 |
| TC (mmol/L) | 4.67 ± 0.89 | 4.58 ± 0.85 | 4.84 ± 0.92 | <0.001 |
| TG (mmol/L) | 1.27 ± 0.93 | 1.13 ± 0.79 | 1.54 ± 1.11 | <0.001 |
| HDL-c (mmol/L) | 1.38 ± 0.31 | 1.40 ± 0.31 | 1.34 ± 0.30 | <0.001 |
| LDL-c (mmol/L) | 2.74 ± 0.67 | 2.68 ± 0.65 | 2.86 ± 0.69 | <0.001 |
| ALT (U/L) | 23.23 ± 21.73 | 21.03 ± 20.77 | 27.47 ± 22.88 | <0.001 |
| BUN (mmol/L) | 4.61 ± 1.17 | 4.53 ± 1.15 | 4.77 ± 1.19 | <0.001 |
| Scr (μmol/L) | 69.66 ± 15.73 | 67.89 ± 15.06 | 73.07 ± 16.42 | <0.001 |
| FPG (mmol/l) | 4.77 ± 0.49 | 4.73 ± 0.49 | 4.84 ± 0.48 | <0.001 |
| Sex | | | | <0.001 |
| Male | 97,778 (53.07%) | 55,336 (45.64%) | 42,442 (67.35%) | |
| Female | 86,474 (46.93%) | 65,901 (54.36%) | 20,573 (32.65%) | |
| Smoking status | | | | <0.001 |
| Current smoker | 9,665 (18.95%) | 5,528 (16.90%) | 4,137 (22.62%) | |
| Ever smoker | 2,116 (4.15%) | 1,227 (3.75%) | 889 (4.86%) | |
| Never smoker | 39,213 (76.90%) | 25,948 (79.34%) | 13,265 (72.52%) | |
| Drinking status | | | | <0.001 |
| Current drinker | 986 (1.93%) | 482 (1.47%) | 504 (2.76%) | |
| Ever drinker | 2,116 (4.15%) | 4,442 (13.58%) | 2,926 (16.00%) | |
| Never drinker | 42,640 (83.62%) | 27,779 (84.94%) | 14,861 (81.25%) | |
| Family history of diabetes | | | | <0.001 |
| No | 180,608 (98.02%) | 118,670 (97.88%) | 61,938 (98.29%) | |
| Yes | 3,644 (1.98%) | 2,567 (2.12%) | 1,077 (1.71%) | |
| Follow-up (years) | 3.14 ± 0.94 | 3.14 ± 0.94 | 3.16 ± 0.94 | 0.004 |
| Prediabetes rate | 20,743 (11.26%) | 10,230 (8.44%) | 10,513 (16.68%) | <0.001 |

Continuous variables were summarized as means (SD) or medians (quartile interval); categorical variables were displayed as percentages (%)

Abbreviations: BMI, body mass index; SBP, systolic blood pressure; DBP; diastolic blood pressure; TC, total cholesterol; TG, triglycerides; HDL-c, high-density lipoprotein cholesterol; LDL-c, low-density lipoprotein cholesterol; ALT, alanine aminotransferase; BUN, blood urea nitrogen; Scr, serum creatinine, FPG, fasting plasma glucose.

normotensive group had a higher HDL-c level (mmol/L), percentage female and family history of diabetes.

Table 2 shows the Clinical and biochemical characteristics of the study participants between the predibetes and no predibetes group. Compared to the no prediabetes group, the prediabetes group had higher age, height, weight, BMI, PP, TC, TG, LDL-c, ALT, BUN, Scr, FPG, SBP, DBP and rate of males, smokers, drinkers, and family history of diabetes.

**Table 2. Clinical and biochemical characteristics between the prediabetes and no prediabetes group.**

| Variable | No prediabetes group | Prediabetes group | P-value |
|---|---|---|---|
| | (n = 163,509) | (n = 20,743) | |
| Age (years) | 40.30 ± 11.70 | 46.72 ± 13.55 | <0.001 |
| Height (cm) | 166.31 ± 8.31 | 167.02 ± 8.36 | <0.001 |
| Weight (kg) | 63.44 ± 11.93 | 68.16 ± 12.08 | <0.001 |
| BMI (kg/m2) | 22.82 ± 3.22 | 24.34 ± 3.32 | <0.001 |
| SBP (mmHg) | 116.98 ± 15.42 | 124.54 ± 17.20 | <0.001 |
| DBP (mmHg) | 73.04 ± 10.41 | 77.35 ± 11.25 | <0.001 |
| PP (mmHg) | 43.93 ± 11.02 | 47.19 ± 12.82 | <0.001 |
| TC (mmol/L) | 4.64 ± 0.88 | 4.85 ± 0.91 | <0.001 |
| TG (mmol/L) | 1.23 ± 0.90 | 1.57 ± 1.15 | <0.001 |
| HDL-c (mmol/L) | 1.38 ± 0.31 | 1.34 ± 0.29 | <0.001 |
| LDL-c (mmol/L) | 2.73 ± 0.67 | 2.84 ± 0.67 | <0.001 |
| ALT (U/L) | 22.66 ± 21.27 | 27.66 ± 24.62 | <0.001 |
| BUN (mmol/L) | 4.58 ± 1.16 | 4.82 ± 1.19 | <0.001 |
| Scr (μmol/L) | 69.27 ± 15.74 | 72.76 ± 15.34 | <0.001 |
| FPG (mmol/l) | 4.73 ± 0.49 | 5.02 ± 0.42 | <0.001 |
| Sex | | | <0.001 |
| Male | 84,403 (51.62%) | 13,375 (64.48%) | |
| Female | 79,106 (48.38%) | 7,368 (35.52%) | |
| Smoking status | | | <0.001 |
| Current smoker | 8,227 (18.23%) | 1,438 (24.56%) | |
| Ever smoker | 1,845 (4.09%) | 271 (4.63%) | |
| Never smoker | 35,067 (77.69%) | 4,146 (70.81%) | |
| Drinking status | | | <0.001 |
| Current drinker | 821 (1.82%) | 165 (2.82%) | |
| Ever drinker | 6,415 (14.21%) | 953 (16.28%) | |
| Never drinker | 37,903 (83.97%) | 4,737 (80.91%) | |
| Family history of diabetes | | | <0.001 |
| No | 160,364 (98.08%) | 20,244 (97.59%) | |
| Yes | 3,145 (1.92%) | 499 (2.41%) | |
| Follow-up (years) | 3.13 ± 0.94 | 3.28 ± 0.95 | <0.001 |

Continuous variables were summarized as means (SD) or medians (quartile interval); categorical variables were displayed as percentages (%)

Abbreviations: BMI, body mass index; SBP, systolic blood pressure; DBP; diastolic blood pressure; TC, total cholesterol; TG, triglycerides; HDL-c, high-density lipoprotein cholesterol; LDL-c, low-density lipoprotein cholesterol; ALT, alanine aminotransferase; BUN, blood urea nitrogen; Scr, serum creatinine, FPG, fasting plasma glucose.

## The incidence rate of prediabetes

Table 3 illustrates the rate of prediabetes among 184,252 individuals during the follow-up period. The overall incidence rate was found to be 11.26% (95%CI: 11.11–11.4%). Specifically, the normotensive group exhibited an incidence rate of 8.44% (95%CI: 8.28–8.59%), whereas the hypertensive group had a significantly higher incidence rate of 16.68% (95%CI: 16.39–16.97%). The cumulative incidence rate for the entire population was calculated to be 358.186 per 10,000 person-years. Within the normotensive group, the cumulative incidence rate was 269.085 per 10,000 person-years, while the hypertensive group demonstrated a notably higher cumulative incidence rate of 528.463 per 10,000 person-years. The statistical analysis revealed

**Table 3. The incidence rate of prediabetes (% or number per 10,000 person-years).**

| Variable | Participants (n) | Prediabetes events (n) | Cumulative incidence rate (95%CI) (%) | Per 10,000 person-years |
|---|---|---|---|---|
| Total | 184,252 | 20,743 | 11.26 (11.11–11.40) | 358.186 |
| Normotension | 121,237 | 10,230 | 8.44 (8.28–8.59) | 269.085 |
| Hypertension | 63,015 | 10,513 | 16.68 (16.39–16.97) | 528.463 |
| P for trend | | | <0.001 | <0.001 |

a significant association between hypertension and an increased incidence rate of prediabetes compared to the normotensive group.

According to the Kaplan-Meier curves presented in Fig 2, the probability of remaining free from prediabetes varied significantly between the two groups (P < 0.0001). As the levels of hypertension increased, the likelihood of avoiding prediabetes gradually decreased. These findings indicate that individuals in the hypertensive group faced a greater risk of developing prediabetes.

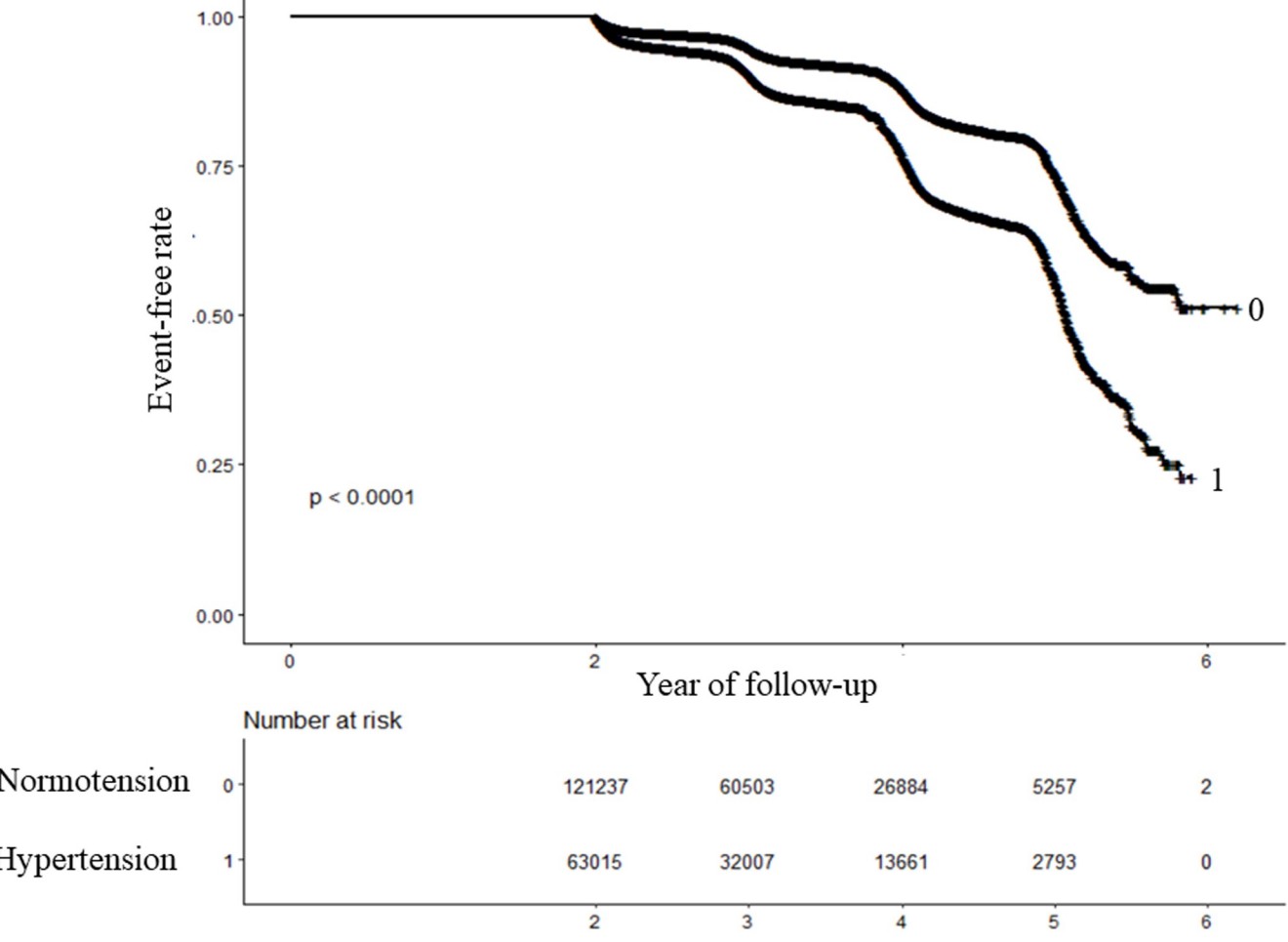

**Fig 2. Kaplan–Meier event-free survival curve.** Kaplan–Meier analysis of incident prediabetes based on two group (log-rank, P < 0.0001).

## Factors influencing the risk of prediabetes determined by univariate analysis

As shown in Table 4, univariate analyses indicated that the PP was positively associated with prediabetes during follow-up (P<0.05). Similar results were found for the subjects' age, male sex, and the BMI, TC, TG, FPG, ALT, BUN, Scr, TG, LDL-c (all P<0.05). The HDL-c was negatively associated with the risk of prediabetes. Men were more prone to developing prediabetes than women. The risk of developing prediabetes was lower for participants who never smoked or drank alcohol. Notably, the risk of prediabetes was not found to be associated with a family history of diabetes (all P>0.05).

## The relationship between the PP and prediabetes

Cox proportional hazard regression models were used to assess the risk, and the HR and 95% confidence interval (CI) for the relationship between the PP (20 mmHg) and prediabetes are shown in Table 5. For hypertensive individuals, in the unadjusted model, the HR for the association between prediabetes and the PP was 1.34, with a 95% CI of 1.31 to 1.38. In model I with adjustments for age and gender, the HR for the association between prediabetes and the PP was 1.19, with a 95% CI of 1.16 to 1.23. In model II with adjustments for the age, sex, BMI, TC, TC, FPG, ALT, BUN, Scr, TG, LDL-c, HDL-c, family history of diabetes, drinking status, and smoking status, the HR for the association between prediabetes and the PP was 1.11, with a 95% CI of 1.03 to 1.19. This finding indicates that for each 20-mmHg increase in the PP, the

**Table 4. Risk of prediabetes analyzed by univariate Cox proportional hazards regression.**

| Variable | Value | HR (95% CI) P-value |
|---|---|---|
| Age (years) | 41.02 ± 12.10 | 1.03 (1.03, 1.03) <0.0001 |
| Sex | | |
| Male | 97,778 (53.07%) | Ref |
| Female | 86,474 (46.93%) | 0.64 (0.62, 0.65) <0.0001 |
| BMI (kg/m$^2$) | 22.99 ± 3.27 | 1.12 (1.12, 1.13) <0.0001 |
| TC (mmol/L) | 4.67 ± 0.89 | 1.22 (1.21, 1.24) <0.0001 |
| TG (mmol/L) | 1.27 ± 0.93 | 1.20 (1.19, 1.21) <0.0001 |
| HDL-c (mmol/L) | 1.38 ± 0.31 | 0.89 (0.84, 0.94) <0.0001 |
| LDL-c (mmol/L) | 2.74 ± 0.67 | 1.27 (1.24, 1.30) <0.0001 |
| ALT (U/L) | 23.23 ± 21.73 | 1.00 (1.00, 1.00) <0.0001 |
| BUN (mmol/L) | 4.61 ± 1.17 | 1.14 (1.13, 1.15) <0.0001 |
| Scr (mmol/L) | 69.66 ± 15.73 | 1.01 (1.01, 1.01) <0.0001 |
| PP (mmHg) | 44.30 ± 11.28 | 1.02 (1.02, 1.03) <0.0001 |
| FPG (mmol/l) | 4.77 ± 0.49 | 5.72 (5.52, 5.92) <0.0001 |
| Smoking status | | |
| Current smoker | 9,665 (18.95%) | 1.0 |
| Ever smoker | 2,116 (4.15%) | 0.79 (0.69, 0.90) 0.0003 |
| Never smoker | 3,9213 (76.90%) | 0.71 (0.67, 0.76) <0.0001 |
| Drinking status | | |
| Current drinker | 986 (1.93%) | 1.0 |
| Ever drinker | 7,368 (14.45%) | 0.63 (0.53, 0.74) <0.0001 |
| Never drinker | 42,640 (83.62%) | 0.58 (0.50, 0.68) <0.0001 |
| Family history of diabetes | | |
| No | 180,608 (98.02%) | 1.0 |
| Yes | 3,644 (1.98%) | 1.04 (0.95, 1.14) 0.3704 |

**Table 5. Relationship between the PP and risk of prediabetes in different models.**

| | Normotension (n = 121,237), HR (95%CI) P-value | | | Hypertension (n = 63,015), HR (95%CI) P-value | | |
|---|---|---|---|---|---|---|
| | Unadjusted | Model 1 | Model 2 | Unadjusted | Model 1 | Model 2 |
| PP (per 20 mmHg increase) | 1.56 (1.49, 1.63) <0.0001 | 1.44 (1.37, 1.50) <0.0001 | 1.06 (0.94, 1.18) 0.3529 | 1.34 (1.31, 1.38) <0.0001 | 1.19 (1.16, 1.23) <0.0001 | 1.11 (1.03, 1.19) 0.0045 |
| PP (quartile) | | | | | | |
| Q1 (≤35) | Ref | Ref | Ref | Ref | Ref | Ref |
| Q2 (36–42) | 1.17 (1.10, 1.24) <0.0001 | 1.15 (1.09, 1.22) <0.0001 | 1.10 (0.95, 1.28) 0.1865 | 1.15 (1.08, 1.22) <0.0001 | 1.10 (1.03, 1.16) 0.0018 | 0.92 (0.80, 1.06) 0.2266 |
| Q3 (43–50) | 1.42 (1.34, 1.50) <0.0001 | 1.35 (1.27, 1.43) <0.0001 | 1.14 (0.98, 1.32) 0.0838 | 1.33 (1.25, 1.41) <0.0001 | 1.23 (1.16, 1.30) <0.0001 | 1.06 (0.93, 1.22) 0.3849 |
| Q4 (≥51) | 1.68 (1.59, 1.78) <0.0001 | 1.54 (1.45, 1.62) <0.0001 | 1.11 (0.96, 1.28) 0.1465 | 1.64 (1.56, 1.73) <0.0001 | 1.38 (1.31, 1.46) <0.0001 | 1.19 (1.04, 1.37) 0.0110 |
| P for trend | <0.0001 | <0.0001 | 0.1599 | <0.0001 | <0.0001 | 0.0017 |

Crude model: No adjustment for other covariates. Model I: Adjusted for age and sex.

Model II: Adjusted for age, sex, BMI, TG, TC, ALT, BUN, Scr, TG, LDL-c, HDL-c, family history of diabetes, drinking status, and smoking status. HR, hazard ratio; CI, confidence interval, Ref, reference.

risk of prediabetes increased by 11% (HR 1.11, 95%CI: 1.03, 1.19, p = 0.0045). We subsequently divided the subjects into four groups based on the PP. We found that the risk of prediabetes in the fourth quartile was 1.19 times higher than that in the first quartile. However, this relationship was not significant in the normotensive individuals (HR 1.06, 95%CI: 0.94, 1.18, p = 0.3529).

## The results of sensitivity analyses

To ensure the robustness of our results, we conducted additional sensitivity analyses. We first performed a sensitivity analysis on individuals with a BMI < 28 kg/m$^2$. After adjusting for confounding factors, we observed a significant positive association between the PP and the risk of prediabetes in hypertensive individuals (HR: 1.11, 95% CI: 1.03–1.20, P = 0.0063). However, this association was not statistically significant in the normotensive individuals (P = 0.5109) (Table 6). We then conducted sensitivity analyses on individuals without a family history of diabetes and those who never consumed alcohol. We obtained similar results for the groups in these analyses (Table 6). Taken together, these sensitivity analyses support the robustness of our findings.

## The nonlinear relationship between the PP and prediabetes in hypertensive individuals

A smooth curve fit in the Cox proportional hazards regression analysis revealed that there is a non-linear relationship between the PP and the risk of prediabetes in hypertensive individuals (Fig 3). A two-piecewise Cox proportional hazards regression model revealed a threshold value of 31 mmHg for the PP (P = 0.033 based on the log-likelihood ratio test). When the PP was <31 mmHg, there was no significant association with the risk of prediabetes (HR: 0.83, 95% CI: 0.59–1.15, P = 0.2646). However, when the PP was ≥31 mmHg, there was a significant positive association with the risk of prediabetes (HR: 1.22, 95% CI: 1.04–2.08, P < 0.0001) (Table 7).

**Table 6. Relationship between the PP and the risk of prediabetes in different sensitivity analyses.**

|  | Normotensive (n = 121,237), HR (95%) P-value | | | Hypertensive (n = 63,015), HR (95%) P-value | | |
|---|---|---|---|---|---|---|
|  | Model 1 | Model 2 | Model 3 | Model 1 | Model 2 | Model 3 |
| PP (per 20 mmHg) | 1.06 (0.94, 1.19) 0.3763 | 1.05 (0.94, 1.18) 0.3843 | 1.04 (0.92, 1.19) 0.5109 | 1.12 (1.04, 1.21) 0.0026 | 1.13 (1.05, 1.21) 0.0012 | 1.11 (1.03, 1.20) 0.0063 |
| PP (quartile) |  |  |  |  |  |  |
| Q1 (≤35) | Ref | Ref | Ref | Ref | Ref | Ref |
| Q2 (36–42) | 1.11 (0.96, 1.30) 0.1627 | 1.11 (0.96, 1.29) 0.1648 | 1.02 (0.87, 1.20) 0.8142 | 0.94 (0.81, 1.10) 0.4434 | 0.93 (0.80, 1.07) 0.3085 | 0.92 (0.78, 1.09) 0.3448 |
| Q3 (43–50) | 1.13 (0.97, 1.31) 0.1201 | 1.14 (0.99, 1.32) 0.0762 | 1.13 (0.97, 1.33) 0.1263 | 1.11 (0.96, 1.30) 0.1644 | 1.08 (0.93, 1.24) 0.3249 | 1.06 (0.90, 1.24) 0.4799 |
| Q4 (≥51) | 1.14 (0.98, 1.32) 0.0935 | 1.11 (0.96, 1.28) 0.1530 | 1.06 (0.91, 1.25) 0.4537 | 1.27 (1.09, 1.47) 0.0017 | 1.23 (1.07, 1.42) 0.0043 | 1.22 (1.04, 1.42) 0.0133 |
| P for trend | 0.1156 | 0.1718 | 0.2847 | 0.0003 | 0.0006 | 0.003 |

**Model I** was a sensitivity analysis performed after excluding participants with a BMI ≥ 28 mmol/L. The model was adjusted for age, sex, BMI, ALT, BUN, Scr, TC, TG, LDL-c, HDL-c, family history of diabetes, drinking status, and smoking status.

**Model II** was a sensitivity analysis performed on participants without a family history of diabetes. The model was adjusted for age, sex, BMI, ALT, BUN, Scr, TC, TG, LDL-c, HDL-c, drinking status, and smoking status.

**Model III** was a sensitivity analysis performed on participants who were never drinkers. The model was adjusted for age, sex, BMI, ALT, BUN, Scr, TC, TG, LDL-c, HDL-c, smoking status, and family history of diabetes.

**HR**, hazard ratio; **CI**, confidence interval, **Ref**, reference.

## Discussion

After adjusting for confounding covariates, we observed a significant positive association between PP and the risk of prediabetes in hypertensive individuals, but not in normotensive individuals. Additionally, we also found a nonlinear association between PP and the risk of prediabetes in hypertensive individuals. A two-piecewise Cox proportional hazards regression model identified a nonlinear inflection point at 31 mmHg.

Hypertension is a significant risk factor for both diabetes and prediabetes [17, 18]. Most studies have traditionally focused on using systolic or diastolic blood pressure as key indicators. However, the PP, which is derived from both systolic and diastolic blood pressure measurements. The PP is primarily influenced by arterial stiffness and has become a widely recognized indirect measure of arterial stiffness [19]. Compared to systolic and diastolic blood pressure, PP has the characteristic of smaller fluctuations and greater stability. And it provides more information about the functional status and vascular elasticity of the cardiovascular system Therefore, PP has garnered increasing attention in recent years. A cohort study involving 200,000 participants showed that the PP is associated with the risk of developing diabetes. They demonstrated that female subjects with a larger PP were more likely to develop diabetes, but this was not observed in male subjects [14]. In addition, an increase in the PP was found to be associated with an increased risk of diabetes complications. Several studies have shown that a high PP may be related to the development of diabetic nephropathy and cardiovascular diseases [20, 21]. Prediabetes is closely associated with the subsequent development of diabetes. Like diabetes, being overweight or obese, hypertensive, and having high cholesterol and/or an advanced age are also risk factors for prediabetes [9, 22]. However, there has been insufficient evidence to establish a clear relationship between the PP and prediabetes in Chinese people. In addition, the effect of hypertension on the relationship between PP and prediabetes needs further study. Therefore, our study analyzed the relationship between PP and prediabetes in detail by analyzing the data of large samples of individuals in multiple regions of China.

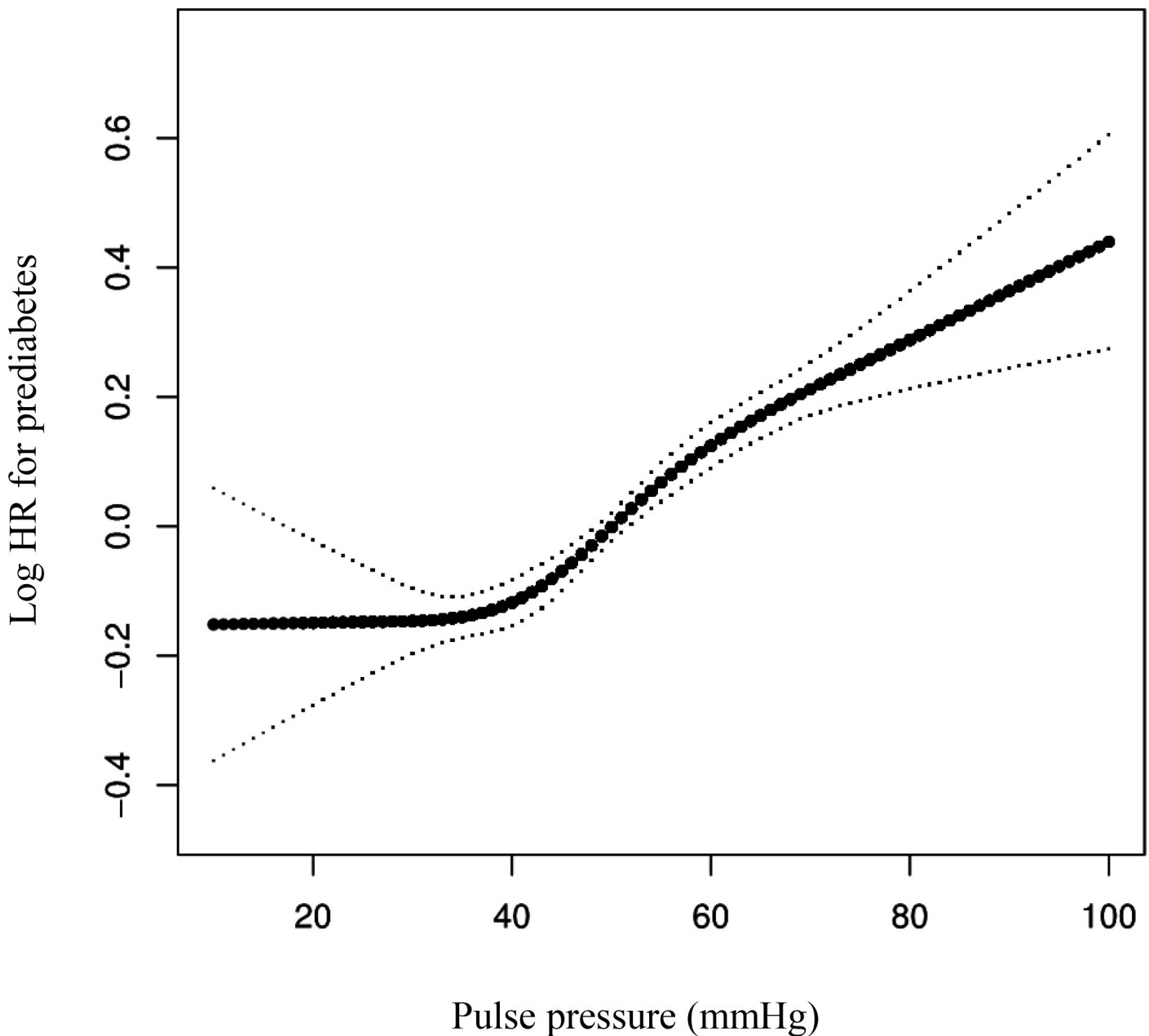

**Fig 3. The nonlinear relationship between PP and risk of prediabetes.** A nonlinear relationship between them was detected after adjusting for gender, age, BMI, TG, TC, HDL-c, LDL-C, ALT, Scr, BUN.

The present study included 184,252 participants without diabetes from 32 regions in China. We confirmed that there is a positive correlation between the PP and the risk of developing prediabetes in our population by multivariate Cox regression analyses. Additionally, we found that a higher PP (Q4) was more likely to lead to prediabetes. Our results suggest that the PP can predict the risk of prediabetes. Interestingly, we also observed that hypertensive subjects had a significantly higher probability of developing prediabetes compared to the normal group (8.44% vs 16.18%). The cumulative incidence rate was 269.085 per 10,000 person-years in the normal group, while the hypertensive group demonstrated a notably higher cumulative

**Table 7. The results of the two-piecewise Cox proportional hazards regression model.**

| Outcome: prediabetes | HR, 95%CI | P-value |
|---|---|---|
| Fitting model by standard Cox regression | 1.20 (1.16, 1.24) | <0.0001 |
| Fitting model by two-piecewise Cox regression | | |
| Inflection point of the PP (20 mmHg) | 31 | |
| <31 mmHg | 0.83 (0.59, 1.15) | 0.2646 |
| ≥31mmHg | 1.22 (1.04, 2.08) | <0.0001 |
| P for the log-likelihood ratio test | | 0.033 |

incidence rate of 528.463 per 10,000 person-years. Kaplan-Meier curves demonstrated that individuals in the hypertensive group had a greater risk of developing prediabetes ($P < 0.0001$). In addition, we found that there was an increased risk of prediabetes associated with a high PP in hypertensive individuals even after adjusting for confounding covariates. However, this association was not observed in individuals with normal blood pressure. The relationship between the PP and prediabetes was nonlinear. Based on a two-piecewise Cox proportional hazards regression model, we calculated the inflection point of the PP. When the PP level was above 31 mmHg, the risk of developing prediabetes increased by 22% for every 20 mmHg increase in the PP. However, when the PP level is below 31 mmHg, there was no association between the PP level and the occurrence of prediabetes. This suggests that in individuals with hypertension, in addition to treating the hypertension, controlling the PP to keep it below 31 mmHg may reduce the risk of prediabetes.

The mechanism by which a high PP increases the risk of prediabetes in hypertensive individuals, but not in normotensive individuals, is not fully understood and may be related to several factors. First, a high PP may reflect arterial stiffness and vascular endothelial dysfunction [23], which can lead to insulin resistance and impaired pancreatic beta-cell function. A high PP may also be associated with increased levels of inflammation and oxidative stress [24], both of which are implicated in the development of prediabetes. Finally, a high PP may be related to dysregulation of the neuroendocrine system, which plays a crucial role in regulating blood glucose metabolism and insulin secretion.

The study's findings have significant clinical implications. Firstly, they provide valuable insights for risk stratification in clinical settings, particularly for hypertensive individuals, aiding healthcare professionals in targeted interventions and personalized preventive measures. Secondly, the identification of individuals with hypertensive conditions and a $PP \geq 31$ mmHg as being at significantly elevated risk of prediabetes suggests the need for early intervention strategies such as lifestyle modifications and proactive management.

However, there were also some potential limitations associated with this study. First, this was a retrospective analysis of a prospective registry study. Second, the diagnosis of prediabetes primarily relies on the oral glucose tolerance test (OGTT), fasting blood glucose, and glycated hemoglobin (HbA1c). Since this study lacked OGTT and HbA1c measurements, we may have incorrectly estimated the incidence of prediabetes. And, the lack of a second blood pressure measurement in the database prevented us from obtaining information about blood pressure variability, which limits our understanding of the impact of various blood pressure indicators on prediabetes. Third, due to the large size of the database, missing data were inevitable. In our analysis, we treated the individuals with missing values for each categorical covariate as a separate group, although there were no significant differences between the groups. Additionally, there are many factors that can influence prediabetes, and this study included limited variables and could not control for other factors such as diet, and exercise [25, 26]. Future research can incorporate more confounding factors. Lastly, while the sample in this study involved a large

population of individuals, further validation is needed for other populations in different countries.

## Conclusion

In conclusion, there is a non-linear relationship between PP and the risk of prediabetes in hypertensive individuals. Maintaining PP within 31 mmHg is crucial for preventing the occurrence of prediabetes. These findings provide insights into the relationship between PP and prediabetes and offer guidance and recommendations for prediabetes prevention.

## Supporting information

**S1 Data.**
(ZIP)

## Acknowledgments

Most of the data and methodology used for this secondary analysis were reported by Chen, Ying et al. (2018). That study's authors deserve our gratitude.

## Author Contributions

**Conceptualization:** Zhanxing Wu, Zhenhua Huang.

**Data curation:** Zhanxing Wu, Zhongqing Chen, Wenfei Zeng, Zhenhua Huang.

**Formal analysis:** Zhanxing Wu, Zhongqing Chen.

**Writing – original draft:** Zhanxing Wu, Zhongqing Chen, Wenfei Zeng, Zhenhua Huang.

**Writing – review & editing:** Ganggang Peng.

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
