## [Decision Letter · Decision Letter 0]

29 Jan 2024

PONE-D-23-37746High pulse pressure is associated with an increased risk of prediabetes in hypertensive individuals: A retrospective study based on an adult Chinese populationPLOS ONE

Dear Dr. Huang,

Thank you for submitting your manuscript to PLOS ONE. After careful consideration, we feel that it has merit but does not fully meet PLOS ONE’s publication criteria as it currently stands. Therefore, we invite you to submit a revised version of the manuscript that addresses the points raised during the review process.

We look forward to receiving your revised manuscript.

Kind regards,

Amirmohammad Khalaji

Academic Editor

PLOS ONE

Journal Requirements:

3. Did you know that depositing data in a repository is associated with up to a 25% citation advantage (https://doi.org/10.1371/journal.pone.0230416)? If you’ve not already done so, consider depositing your raw data in a repository to ensure your work is read, appreciated and cited by the largest possible audience. You’ll also earn an Accessible Data icon on your published paper if you deposit your data in any participating repository (https://plos.org/open-science/open-data/#accessible-data).

Reviewers' comments:

Reviewer's Responses to Questions

**Comments to the Author**

1. Is the manuscript technically sound, and do the data support the conclusions?

Reviewer #1: Yes

Reviewer #2: Yes

Reviewer #3: Yes

2. Has the statistical analysis been performed appropriately and rigorously? 

Reviewer #1: Yes

Reviewer #2: Yes

Reviewer #3: Yes

3. Have the authors made all data underlying the findings in their manuscript fully available?

Reviewer #1: No

Reviewer #2: No

Reviewer #3: Yes

4. Is the manuscript presented in an intelligible fashion and written in standard English?

Reviewer #1: Yes

Reviewer #2: Yes

Reviewer #3: Yes

5. Review Comments to the Author

Reviewer #1: Wu et al. have conducted a retrospective study on the association between high pulse pressure and prediabetes in patients with hypertension. The study findings are interesting and the methodology used is rigorous. There are some comments to address:

- While the first paragraph of the discussion should focus on the main findings, it should be stated in a different way from the results. For instance, the numbers and p-values should not be restated.

- In the introduction section, the gap in knowledge and the rationale should be stated in more detail.

- It is highly suggested to have a Table presenting the demographic characteristics of those with and without prediabetes as well.

Reviewer #2: The study investigates the relationship between pulse pressure (PP) and prediabetes, focusing on its association with hypertension. Analyzing data from 184,252 Chinese adults, the study reveals a positive link between PP and prediabetes in hypertensive individuals. A non-linear relationship is observed, with an inflection point at 31 mmHg.

Despite the potential interest in the study, I have some important concerns that I have discussed below. these whole sections should be revised:

In the introduction, the references should be updated, particularly regarding the global prevalence of prediabetes, with the current reference dating back to 2012. Consider incorporating recent statistics and additional references to enrich this section. Regarding the statement "PP may be associated with greater arterial stiffness," it's advisable to rephrase since it's now widely accepted that PP is indeed linked to arterial stiffness. This potential error might be attributed to the reliance on outdated references. In the clinical practice discussion, there's a mention of blood pressure being associated with prediabetes without a reference. To support this claim, include a reference and delve into the superiority of using PP over blood pressure. Elaborate on the clinical implications of the research findings.

In the methods section, it's essential to clarify whether any inclusion and exclusion criteria were applied during participant selection. Clearly outline the exclusion criteria, such as individuals with a history of diseases like coronary artery disease, hyperlipidemia, or autoimmune conditions requiring corticosteroid treatment.

Concerning the results, some paragraphs need improvement in clarity, such as the one mentioning "percentage female." Professional proof-editing can enhance these sections.

In the discussion, address the limited and insufficient references. Add relevant references and explain why there's been an increased focus on PP in recent years. Comprehensive revisions are necessary to address the clinical implications of the findings.

In general, a notable aspect of this research lies in establishing a PP cutoff point at 31 mmHg. However, the study requires substantial revisions for structural and clarity issues, and additional discussion is needed to explore the clinical implications of the findings.

Reviewer #3: Wu et al used an observational study design to investigate the relationship of pulse pressure with incident pre-diabetes (i.e. IFG) in a retrospective study of a Chinese population. They discovered a significant positive relationship in hypertensive individuals, but none in normotensive individuals. There are pre-existing data on PP and incident diabetes, so the novelty in this study is the relationship with pre-diabetes and using a different ethnic group. The large sample size is the defining attribute of this study which also allowed them to determine that a PP threshold of 31 mm Hg in hypertensive individuals. Their methodologies are reasonably well-described and robust. They adequately described their limitations of their study and offered reasonable hypotheses to describe their findings.

A few minor comments:

Line 68: "70% of individuals" probably includes individuals with IGT and not only IFG. Please check the reference.

Line 110: The authors stated they did not need ethical approval may be correct, but it would better if the authors specified that the participants were anonymised before analysis.

Line 127: Were the staff retrained over the years, and was their inter- and intra- variability determined?

Line 128: Were assays all done at the same laboratory?

Line 167: The authors had a fully adjusted model including BMI in their results section, but BMI is not stated in the statistical analysis

6. PLOS authors have the option to publish the peer review history of their article (what does this mean?). If published, this will include your full peer review and any attached files.

Reviewer #1: No

Reviewer #2: **Yes: **Tara Reshadmanesh

Reviewer #3: No

---

## [Author Response · Author response to Decision Letter 0]

14 Feb 2024

Reviewer’s questions and suggestions

Reviewer #1: Wu et al. have conducted a retrospective study on the association between high pulse pressure and prediabetes in patients with hypertension. The study findings are interesting and the methodology used is rigorous. There are some comments to address:

1) While the first paragraph of the discussion should focus on the main findings, it should be stated in a different way from the results. For instance, the numbers and p-values should not be restated.

Respond: We agree with this suggestion and have modified the terminology in this paragraph. Here is the corrected representation of the modified first paragraph: After adjusting for confounding covariates, we observed a significant positive association between PP and the risk of prediabetes in hypertensive individuals, but not in normotensive individuals. Additionally, we also found a nonlinear association between PP and the risk of prediabetes in hypertensive individuals. A two-piecewise Cox proportional hazards regression model identified a nonlinear inflection point at 31 mmHg.

2) In the introduction section, the gap in knowledge and the rationale should be stated in more detail.

Respond: Thank you for your suggestion. I have added content and new references in the introduction section to enhance the overall logical structure of the article.

3）It is highly suggested to have a Table presenting the demographic characteristics of those with and without prediabetes as well.

Respond: We agree with this suggestion and have added table 2. After the addition of Table 2, we observed an enrichment of the article’s content, rendering the entire paper more comprehensive and logically coherent.We will be happy to edit the text further, based on helpful comments from the reviewers.

TABLE 2. Clinical and biochemical characteristics between the predibetes and no predibetes group.

Variable No prediabetes

(n = 163,509) Prediabtes group

 (n = 20,743) P-value

Age (years) 40.30 ± 11.70 46.72 ± 13.55 <0.001

Height (cm) 166.31 ± 8.31 167.02 ± 8.36 <0.001

Weight (kg) 63.44 ± 11.93 68.16 ± 12.08 <0.001

BMI (kg/m2) 22.82 ± 3.22 24.34 ± 3.32 <0.001

SBP (mmHg) 116.98 ± 15.42 124.54 ± 17.20 <0.001

DBP (mmHg) 73.04 ± 10.41 77.35 ± 11.25 <0.001

PP (mmHg) 43.93 ± 11.02 47.19 ± 12.82 <0.001

TC (mmol/L) 4.64 ± 0.88 4.85 ± 0.91 <0.001

TG (mmol/L) 1.23 ± 0.90 1.57 ± 1.15 <0.001

HDL-c (mmol/L) 1.38 ± 0.31 1.34 ± 0.29 <0.001

LDL-c (mmol/L) 2.73 ± 0.67 2.84 ± 0.67 <0.001

ALT (U/L) 22.66 ± 21.27 27.66 ± 24.62 <0.001

BUN (mmol/L) 4.58 ± 1.16 4.82 ± 1.19 <0.001

Scr (μmol/L) 69.27 ± 15.74 72.76 ± 15.34 <0.001

FPG (mmol/l) 4.73 ± 0.49 5.02 ± 0.42 <0.001

Sex <0.001

Male 84,403 (51.62%) 13,375 (64.48%) 

Female 79,106 (48.38%) 7,368 (35.52%) 

Smoking status <0.001

Current smoker 8,227 (18.23%) 1,438 (24.56%) 

Ever smoker 1,845 (4.09%) 271 (4.63%) 

Never smoker 35067 (77.69%) 4146 (70.81%) 

Drinking status <0.001

Current drinker 821 (1.82%) 165 (2.82%) 

Ever drinker 6,415 (14.21%) 953 (16.28%) 

 Never drinker 37,903 (83.97%) 4,737 (80.91%) 

Family history of diabetes <0.001

No 160,364 (98.08%) 20,244 (97.59%) 

Yes 3,145 (1.92%) 499 (2.41%) 

Follow-up (years) 3.13 ± 0.94 3.28 ± 0.95 <0.001

Continuous variables were summarized as means (SD) or medians (quartile interval); categorical variables were displayed as percentages (%)

Abbreviations: BMI, body mass index; SBP, systolic blood pressure; DBP; diastolic blood pressure; TC, total cholesterol; TG, triglycerides; HDL-c, high-density lipoprotein cholesterol; LDL-c, low-density lipoprotein cholesterol; ALT, alanine aminotransferase; BUN, blood urea nitrogen; Scr, serum creatinine, FPG, fasting plasma glucose.

Reviewer #2: The study investigates the relationship between pulse pressure (PP) and prediabetes, focusing on its association with hypertension. Analyzing data from 184,252 Chinese adults, the study reveals a positive link between PP and prediabetes in hypertensive individuals. A non-linear relationship is observed, with an inflection point at 31 mmHg.

Despite the potential interest in the study, I have some important concerns that I have discussed below. these whole sections should be revised:

1）In the introduction, the references should be updated, particularly regarding the global prevalence of prediabetes, with the current reference dating back to 2012. Consider incorporating recent statistics and additional references to enrich this section.

Respond: We highly appreciate your suggestion. Following your advice, I have incorporated additional and recent content, primarily sourced from the AMA. 2023 , 329(14):1206-1216. After these modifications, the article has become more enriched and comprehensive. 

 2）Regarding the statement "PP may be associated with greater arterial stiffness," it's advisable to rephrase since it's now widely accepted that PP is indeed linked to arterial stiffness. This potential error might be attributed to the reliance on outdated references.

Respond: Thank you for your suggestion. I have made modifications to the content and replaced the references based on your suggestions. The details are as follows：“Arterial stiffness leads to an increased forward wave amplitude and earlier arrival of the reflected wave, resulting in elevated pulse pressure. This indicates a close correlation between PP and arterial stiffness” Reference: Niiranen TJ, Kalesan B, Mitchell GF, Vasan RS. Relative Contributions of Pulse Pressure and Arterial Stiffness to Cardiovascular Disease. Hypertension. 2019 , 73(3):712-717. doi: 10.1161/HYPERTENSIONAHA.118.12289. PMID: 30661478;

3） In the clinical practice discussion, there's a mention of blood pressure being associated with prediabetes without a reference. To support this claim, include a reference and delve into the superiority of using PP over blood pressure. Elaborate on the clinical implications of the research findings. 

Respond: Thank you very much for your suggestion. We believe that the section “The association between blood pressure and prediabetes” is not significant in the article; therefore, I have removed it. In addition, I have added information on the superiority of pulse pressure (PP) over systolic blood pressure. Furthermore, in the discussion section, I have highlighted the clinical significance of our study. The specific details are as follows：

（1） The PP is primarily influenced by arterial stiffness and has become a widely recognized indirect measure of arterial stiffness. Compared to systolic and diastolic blood pressure, PP has the characteristic of smaller fluctuations and greater stability. Therefore, PP has garnered increasing attention in recent years.

（2） The study’s findings have significant clinical implications. Firstly, they provide valuable insights for risk stratification in clinical settings, particularly for hypertensive individuals, aiding healthcare professionals in targeted interventions and personalized preventive measures. Secondly, the identification of individuals with hypertensive conditions and a PP ≥ 31 mmHg as being at significantly elevated risk of prediabetes suggests the need for early intervention strategies such as lifestyle modifications and proactive management.

3）In the methods section, it's essential to clarify whether any inclusion and exclusion criteria were applied during participant selection. Clearly outline the exclusion criteria, such as individuals with a history of diseases like coronary artery disease, hyperlipidemia, or autoimmune conditions requiring corticosteroid treatment.

Respond: Thank you for your question. In the methods section, we have added the inclusion and exclusion criteria. 

The specific content is as follows:

Participants were included at baseline in the original study based on the following criteria: 1) Aged 20 and older who underwent at least two health examinations. 2)Participants with follow-up fasting plasma glucose levels between 6.1 and 6.9 mmol/l and no new reports of diabetes were included in the study.

Participants were excluded at baseline in the original study based on the following criteria: (1) lack of available information on weight, height, and gender; (2) extreme BMI values (<15kg/m2 or >55kg/m2); (3) visit intervals <2 years; (4) no available fasting plasma glucose value; (5) participants diagnosed with diabetes at baseline and participants with undefined diabetes status at follow-up. After applying these exclusion criteria, a total of 211,833 participants remained in the original study. 

In the subsequent analysis, 27,631 participants were further excluded, including those with: 1) missing available FPG values during follow-up, 2) missing available DBP or SBP values, 3) individuals with FPG≥5.6 mmol/l at baseline, 4) individuals with FPG＞6.9 mmol/l during follow-up, and 5) individuals diagnosed as undefined with diabetes during follow-up. A total of 184,252 participants were finally included in this study. The participant selection process is illustrated in Figure 1.

However, our original data did not mention a history of coronary artery disease, hyperlipidemia, or autoimmune diseases requiring corticosteroid treatment. Therefore, the exclusion criteria in this study did not include individuals with a history of coronary artery disease, hyperlipidemia, or autoimmune diseases requiring corticosteroid treatment.

4）Concerning the results, some paragraphs need improvement in clarity, such as the one mentioning "percentage female." Professional proof-editing can enhance these sections.

Respond: Thank you for your question. I have organized the results section to make the paragraphs clearer.

5）In the discussion, address the limited and insufficient references. Add relevant references and explain why there's been an increased focus on PP in recent years. Comprehensive revisions are necessary to address the clinical implications of the findings.

Respond: Thank you for your suggestion. Pulse pressure (PP) values are primarily influenced by arterial stiffness. In comparison to systolic and diastolic blood pressure, PP exhibits minimal fluctuations and greater stability. It has become widely recognized as an indirect measure of arterial stiffness, with research indicating its potential role in glucose metabolism disorders, metabolic syndrome, and insulin resistance. Consequently, PP has garnered increasing attention in recent years. Additionally, we have addressed the inadequacies by incorporating relevant literature. Furthermore, in the discussion section, we have expanded on the clinical significance of our findings. The specific details are as follows：

Firstly, they provide valuable insights for risk stratification in clinical settings, particularly for hypertensive individuals, aiding healthcare professionals in targeted interventions and personalized preventive measures. Secondly, the identification of individuals with hypertensive conditions and a PP ≥ 31 mmHg as being at significantly elevated risk of prediabetes suggests the need for early intervention strategies such as lifestyle modifications and proactive management.

6）In general, a notable aspect of this research lies in establishing a PP cutoff point at 31 mmHg. However, the study requires substantial revisions for structural and clarity issues, and additional discussion is needed to explore the clinical implications of the findings.

Respond: Thank you for your suggestion. I have made revisions to the article and introduced the clinical significance. The study’s findings have significant clinical implications. Firstly, they provide valuable insights for risk stratification in clinical settings, particularly for hypertensive individuals, aiding healthcare professionals in targeted interventions and personalized preventive measures. Secondly, the identification of individuals with hypertensive conditions and a PP ≥ 31 mmHg as being at significantly elevated risk of prediabetes suggests the need for early intervention strategies such as lifestyle modifications and proactive management.

Reviewer #3: Wu et al used an observational study design to investigate the relationship of pulse pressure with incident pre-diabetes (i.e. IFG) in a retrospective study of a Chinese population. They discovered a significant positive relationship in hypertensive individuals, but none in normotensive individuals. There are pre-existing data on PP and incident diabetes, so the novelty in this study is the relationship with pre-diabetes and using a different ethnic group. The large sample size is the defining attribute of this study which also allowed them to determine that a PP threshold of 31 mm Hg in hypertensive individuals. Their methodologies are reasonably well-described and robust. They adequately described their limitations of their study and offered reasonable hypotheses to describe their findings.

A few minor comments:

1）Line 68: "70% of individuals" probably includes individuals with IGT and not only IFG. Please check the reference.

Respond: We found in the article that “the identification of prediabetic individuals is based on impaired fasting glucose, impaired glucose tolerance, and glycated hemoglobin.” Therefore, the “70% of individuals” includes those with IGT and IFG.

2）Line 110: The authors stated they did not need ethical approval may be correct, but it would better if the authors specified that the participants were anonymised before analysis.

Respond: Thank you for your question. I have added a statement about anonymizing participants in the ethical declaration section.

3）Line 127: Were the staff retrained over the years, and was their inter- and intra- variability determined?

Respond: Thank you for your inquiry. Regarding the demographic information and related measurements such as blood pressure, the trained staff underwent retraining to ensure consistency in their work.

4）Line 128: Were assays all done at the same laboratory?

Respond: The original researchers retrospectively obtained data for the Dryad Digital Repository from a computerized database developed by the Rich Healthcare Group in China. This database includes the medical records of individuals who underwent health check-ups between 2010 and 2016, covering 32 regions and 11 cities in China. Therefore, all the tests were conducted in the same laboratory.

5）Line 167: The authors had a fully adjusted model including BMI in their results section, but BMI is not stated in the statistical analysis

Respond: Thank you for your inquiry. In the methodology section, we mentioned that demographic information includes patient height and weight. BMI is calculated based on height and weight.

---

## [Decision Letter · Decision Letter 1]

4 Mar 2024

PONE-D-23-37746R1High pulse pressure is associated with an increased risk of prediabetes in hypertensive individuals: A retrospective study based on an adult Chinese populationPLOS ONE

Dear Dr. Huang,

Thank you for submitting your manuscript to PLOS ONE. After careful consideration, we feel that it has merit but does not fully meet PLOS ONE’s publication criteria as it currently stands. Therefore, we invite you to submit a revised version of the manuscript that addresses the points raised during the review process.

We look forward to receiving your revised manuscript.

Kind regards,

Amirmohammad Khalaji

Academic Editor

PLOS ONE

Journal Requirements:

Reviewers' comments:

Reviewer's Responses to Questions

**Comments to the Author**

1. If the authors have adequately addressed your comments raised in a previous round of review and you feel that this manuscript is now acceptable for publication, you may indicate that here to bypass the “Comments to the Author” section, enter your conflict of interest statement in the “Confidential to Editor” section, and submit your "Accept" recommendation.

Reviewer #1: All comments have been addressed

Reviewer #2: All comments have been addressed

Reviewer #3: (No Response)

2. Is the manuscript technically sound, and do the data support the conclusions?

Reviewer #1: (No Response)

Reviewer #2: Yes

Reviewer #3: Yes

3. Has the statistical analysis been performed appropriately and rigorously? 

Reviewer #1: (No Response)

Reviewer #2: Yes

Reviewer #3: Yes

4. Have the authors made all data underlying the findings in their manuscript fully available?

Reviewer #1: (No Response)

Reviewer #2: Yes

Reviewer #3: Yes

5. Is the manuscript presented in an intelligible fashion and written in standard English?

Reviewer #1: (No Response)

Reviewer #2: Yes

Reviewer #3: Yes

6. Review Comments to the Author

Reviewer #1: (No Response)

Reviewer #2: (No Response)

Reviewer #3: Authors' Response to Comment 1

"We found in the article that “the identification of prediabetic individuals is based on impaired fasting glucose, impaired glucose tolerance, and glycated hemoglobin.” Therefore, the “70% of individuals” includes those with IGT and IFG."

Having clarified that point, the authors need to amend the statement in the introduction to words like "70% of individuals with prediabetes (as defined by IFG and IGT)

Authors' Response to Comment 3

"Regarding the demographic information and related measurements such as blood pressure, the trained staff underwent retraining to ensure consistency in their work."

The authors therefore need to amend the manuscript to include this point. Ensuring consistency ideally means their inter- and intra-rater reliabilities are low. They should include that data if they have it.

Authors' Response to Comment 4

"Therefore, all the tests were conducted in the same laboratory."

Please include this in the methods section.

Authors' Response to Comment 5

" In the methodology section, we mentioned that demographic information includes patient height and weight. BMI is calculated based on height and weight."

The authors need to state in line 187 that Model II included BMI.

7. PLOS authors have the option to publish the peer review history of their article (what does this mean?). If published, this will include your full peer review and any attached files.

Reviewer #1: No

Reviewer #2: **Yes: **Tara Reshadmanesh

Reviewer #3: No

---

## [Author Response · Author response to Decision Letter 1]

11 Mar 2024

2024-3-10 Reviewers’ questions and suggestions

1. Reviewer #3: Authors' Response to Comment 1

"We found in the article that “the identification of prediabetic individuals is based on impaired fasting glucose, impaired glucose tolerance, and glycated hemoglobin.” Therefore, the “70% of individuals” includes those with IGT and IFG." Having clarified that point, the authors need to amend the statement in the introduction to words like "70% of individuals with prediabetes (as defined by IFG and IGT).

Respond: I strongly agree with your perspective and have included this content in the introduction. (line 71-73)

2. Authors' Response to Comment 3

"Regarding the demographic information and related measurements such as blood pressure, the trained staff underwent retraining to ensure consistency in their work."

The authors therefore need to amend the manuscript to include this point. Ensuring consistency ideally means their inter- and intra-rater reliabilities are low. They should include that data if they have it.

Respond: I strongly agree with your perspective and have included this content in the method. (line 125-127)

3. Authors' Response to Comment 4

"Therefore, all the tests were conducted in the same laboratory."

Please include this in the methods section.

Respond: Thank you for your suggestion. We have included this content in the method. (line 131)

4. Authors' Response to Comment 5

" In the methodology section, we mentioned that demographic information includes patient height and weight. BMI is calculated based on height and weight."

The authors need to state in line 187 that Model II included BMI.

Respond: Thank you for your suggestion. We have included this content in the method.

(line 125, and line 167)

---

## [Decision Letter · Decision Letter 2]

25 Mar 2024

High pulse pressure is associated with an increased risk of prediabetes in hypertensive individuals: A retrospective study based on an adult Chinese population

PONE-D-23-37746R2

Dear Dr. Huang,

We’re pleased to inform you that your manuscript has been judged scientifically suitable for publication and will be formally accepted for publication once it meets all outstanding technical requirements.

Kind regards,

Amirmohammad Khalaji

Academic Editor

PLOS ONE

Additional Editor Comments (optional):

Reviewers' comments:

Reviewer's Responses to Questions

**Comments to the Author**

1. If the authors have adequately addressed your comments raised in a previous round of review and you feel that this manuscript is now acceptable for publication, you may indicate that here to bypass the “Comments to the Author” section, enter your conflict of interest statement in the “Confidential to Editor” section, and submit your "Accept" recommendation.

Reviewer #3: All comments have been addressed

2. Is the manuscript technically sound, and do the data support the conclusions?

Reviewer #3: Yes

3. Has the statistical analysis been performed appropriately and rigorously? 

Reviewer #3: Yes

4. Have the authors made all data underlying the findings in their manuscript fully available?

Reviewer #3: Yes

5. Is the manuscript presented in an intelligible fashion and written in standard English?

Reviewer #3: Yes

6. Review Comments to the Author

Reviewer #3: Thank you for your resubmission.

The authors have satisfactorily amended the manuscript with my suggestions.

7. PLOS authors have the option to publish the peer review history of their article (what does this mean?). If published, this will include your full peer review and any attached files.

Reviewer #3: No

---

## [Editor Report · Acceptance letter]

21 Aug 2024

PONE-D-23-37746R2 

PLOS ONE

Dear Dr. Huang, 

I'm pleased to inform you that your manuscript has been deemed suitable for publication in PLOS ONE. Congratulations! Your manuscript is now being handed over to our production team.

Kind regards, 

on behalf of

Dr. Amirmohammad Khalaji 

Academic Editor

PLOS ONE